# Targeted Whole Genome Sequencing (TWG-Seq) of Cucumber Green Mottle Mosaic Virus Using Tiled Amplicon Multiplex PCR and Nanopore Sequencing

**DOI:** 10.3390/plants11202716

**Published:** 2022-10-14

**Authors:** Joanne Mackie, Wycliff M. Kinoti, Sumit I. Chahal, David A. Lovelock, Paul R. Campbell, Lucy T. T. Tran-Nguyen, Brendan C. Rodoni, Fiona E. Constable

**Affiliations:** 1School of Applied Systems Biology, La Trobe University, Melbourne, VIC 3083, Australia; 2Agriculture Victoria Research, Department of Jobs, Precincts and Regions, AgriBio, Melbourne, VIC 3083, Australia; 3Horticulture and Forestry Science, Department of Agriculture and Fisheries, Ecosciences Precinct, Brisbane, QLD 4102, Australia; 4Plant Health Australia, Canberra, ACT 2600, Australia

**Keywords:** CGMMV, tiled amplicon sequencing, nanopore

## Abstract

Rapid and reliable detection tools are essential for disease surveillance and outbreak management, and genomic data is essential to determining pathogen origin and monitoring of transmission pathways. Low virus copy number and poor RNA quality can present challenges for genomic sequencing of plant viruses, but this can be overcome by enrichment of target nucleic acid. A targeted whole genome sequencing (TWG-Seq) approach for the detection of cucumber green mottle mosaic virus (CGMMV) has been developed where overlapping amplicons generated using two multiplex RT-PCR assays are then sequenced using the Oxford Nanopore MinION. Near complete coding region sequences were assembled with ≥100× coverage for infected leaf tissue dilution samples with RT-qPCR cycle quantification (Cq) values from 11.8 to 38 and in seed dilution samples with Cq values 13.8 to 27. Consensus sequences assembled using this approach showed greater than 99% nucleotide similarity when compared to genomes produced using metagenomic sequencing. CGMMV could be confidently detected in historical seed isolates with degraded RNA. Whilst limited access to, and costs associated with second-generation sequencing platforms can influence diagnostic outputs, the portable Nanopore technology offers an affordable high throughput sequencing alternative when combined with TWG-Seq for low copy or degraded samples.

## 1. Introduction

The highly transmissible, cucurbit-infecting cucumber green mottle mosaic virus (CGMMV) was reported for the first time in Australia in 2014. Field crops of watermelon in the Northern Territory (NT) were affected by severe mosaic and stunted foliar development [1,2]. During surveillance more infections were found in surrounding cucurbit crops and weeds and led to infested farms being placed under quarantine control [3]. In the period 2014 to 2020, further CGMMV outbreaks occurred in four Australian states and one territory, and the virus has been detected in *Citrullus lanatus* (watermelon), *Cucumis sativus* (cucumber), *Amaranthus viridis* (Amaranth) *Chenopodium album* (Fat Hen) *Portulaca oleracea* (Pigweed) *and Solanum nigrum* (Black Nightshade) [4,5]. Enzyme linked immunosorbent assays (ELISA) and reverse-transcription polymerase chain reaction (RT-PCR) methods were used to detect CGMMV, assist with tracing and help delimit the outbreaks. Metagenomic high throughput sequencing (HTS) showed that isolates from the Western Australian (WA) outbreak of CGMMV were grouped with an isolate from the NT and inferred a single recent incursion into the affected WA region [6]. The source of most Australian outbreaks has not been determined, although seed is the most likely source of introduction into Australia [7] and subsequent spread is also likely to have occurred.

Australian cucurbit production is heavily reliant on imported seed and subsequent outbreaks have led to the introduction of strict quarantine measures for the importation of seeds of eight cucurbit species (and their hybrids) by the Australian Government Department of Agriculture, Fisheries and Forestry (DAFF). These emergency measures require that all seed, of listed species for import, be tested for CGMMV by ELISA on a sample size of up to 9400 seeds with sub-samples of no more than 100 seeds to ensure adequate sensitivity for CGMMV detection [7]. While a positive detection by ELISA is enough to restrict entry of seed into Australia, confirmation of CGMMV contamination by endpoint reverse transcription-polymerase chain reactions (RT-PCR) with Sanger sequencing of the PCR product provides confidence in a diagnostic result that the pathogen is truly present, and detection is not an off-target reaction to a non-target protein or nucleic acid. This workflow completes an identification of CGMMV but provides limited epidemiological information based on a partial genomic sequence and is confounded by low titre samples. 

Management of plant disease outbreaks and associated surveillance is enhanced by quick, accurate and cost-effective pathogen identification, and understanding an organism’s genetic lineage, through whole genome sequencing, can aid traceability to identify the source and transmission pathways and inform necessary quarantine measures to combat establishment and further spread [8,9,10,11,12]. The use of untargeted or shotgun metagenomic HTS can yield full genomes of viruses present in a sample [12] and support strain-specific identification [13]. Short-read sequencing such as that performed by Illumina sequencing technologies, generates abundant high quality data and can analyse a substantial number of samples in a single run [14]. The requirement for high-level computing equipment and storage, complex library preparation, suitable expertise to perform bioinformatics analysis and lengthy turn-around-time, can be limiting factors when employing this technology for outbreak management [13,15]. Metagenomic HTS for plant viruses using long read nanopore sequencing using a Flongle (Oxford Nanopore Technologies; ONT) has been shown to be a simple and cost effective broad based and untargeted method for virus discovery which can provide results in 4–5 days [16]. However, host dependent variability in sequence coverage was observed, therefore it may not always provide the genomic resolution required to enable pathogen traceability during an outbreak. The use of metagenomic sequencing for plant virus detection is thwarted by an overabundance of host reads, and in samples with low virus titre, sensitivity is often unsatisfactory.

Enrichment methods are frequently employed to raise confidence in sequencing outcomes [11,17,18,19]. Indirect methods that can increase the relative virus titre of a metagenomic sample involve rRNA host depletion, and dsRNA and siRNA extractions [20]. Direct methods are often less expensive, faster, and easier to perform and include target enrichment sequencing, rolling-circle amplification, and PCR amplicon enrichment. Target enrichment sequencing incorporates overlapping capture probes that hybridize to complementary sequences, is prepared in a single tube per sample, and requires fewer PCR cycles than PCR amplification enrichment, however high costs, technical expertise, and time to develop new probe schemes impact implementation for an emerging outbreak [21]. Rolling-circle amplification is a simple and effective enrichment method for circular DNA or RNA [22]. Advantages of this technique are low temperature amplification and generation of long DNA fragments; however linear targets require ligation and RNA amplification is challenging [22]. 

An early application of PCR amplification sequencing, known as RNA ‘jackhammering’ was developed to amplify degraded human immunodeficiency virus (HIV) archive samples, where RNA quality was below limits of quantification and only fragmented virus particles survived [23]. Panels of primers were designed to amplify two sets of short overlapping fragments across the complete coding regions of the virus genome, with each set of amplicons overlapping between but not within each set. Adoption of this tiled amplicon approach combined with long read nanopore sequencing of the amplicons using a MinION (Oxford Nanopore Technology) instrument has enabled the real-time genomic sequencing of human viruses causing infectious diseases such a Zika virus, Dengue virus and severe acute respiratory syndrome coronavirus 2 (SARS-CoV-2) and in turn informed genomic epidemiological analysis for surveillance and transmission management [24,25,26]. The method is highly specific, easy to perform, and primer schemes can be rapidly updated and implemented in response to disease incursions. Whole genomes, multiple gene regions or multiple organisms can be targeted. This portable and low-cost long-read sequencing system is easy to implement and has made HTS accessible to more laboratories. It is capable of producing consensus sequence accuracy of greater than 99% [27,28] which is critical for genomic epidemiology used for traceability [24]. 

In this study, we evaluated a targeted whole genome sequencing (TWG-Seq) approach that incorporated two-reaction multiplex tiling PCR with 7 and 6 primer pairs for amplification and sequencing of the ~6.2 kb coding region of CGMMV using the MinION instrument. Two sample panels comprising serial dilutions of RNA extracted from leaf tissues and seed were used to compare the targeted multiplex PCR tiled amplicons sequenced using the MinION and metagenomic sequencing using an Illumina NovaSeq instrument, assessing coverage and depth across the coding sequence of the genome. CGMMV positive seed interception samples were also tested using both tiled amplicon and metagenomic sequencing to trial the technique on virus-positive samples of unknown copy number and RNA quality. Assay specificity was tested using closely related virus isolates of kyuri green mottle mosaic virus (KGMMV) and zucchini green mottle mosaic virus (ZGMMV). 

## 2. Results

### 2.1. Relative Quantification of Viral Load

A synthetic CGMMV fragment (538 nt) containing binding sites for primers and probes for the RT-qPCR [29] (Appendix A) was used to generate a transcribed RNA standards panel. The panel used in the RT-qPCR [29] produced a standard curve (R2 = 1, Slope = −3.598) that was used to quantify virus copies present in the RNA extracted from the seed and plant panel samples (Appendix A). Quantitative RT-PCR produced cycle quantification (Cq) values that ranged from 9.5 to 38.0 for the plant panel RNA extracted from the undiluted positive leaf tissue homogenate to a dilution in healthy homogenate equivalent to 1.00E-11 dilution (Table 1), and 8.4 to 30.4 for the seed panel samples for RNA extracted from the undiluted homogenate used to contaminate the seeds to the RNA extract from the 1 contaminated seed in 1000 seeds (Table 2). The CGMMV assay efficiency, based on the slope value was determined to be 89.641%. The equivalent CGMMV copy number for the plant panel was 725 million copies per µL in RNA extracted from an undiluted homogenate to 8 virus copies per µL RNA extracted from a dilution of 10^−11^. The equivalent CGMMV copy number for the seed panel was 1526 million copies per µL RNA extracted from a single contaminated seed to 947 copies per µL RNA extracted from one contaminated seed in 1000 (Appendix A).

### 2.2. Validation of the Multiplex Tiling PCR Assays

Using the web-based primer design tool Primal Scheme (https://primalscheme.com (accessed on 28 April 2021)) [25] 13 sets of primers were designed to generate amplicons of approximately 600 nt in length with a 100 nt overlap between adjacent amplicons, across the coding sequence regions of CGMMV (6377 nt) (Appendix A). Each primer pair was tested in a single-plex 2-step RT-PCR using CGMMV isolates VPRI43306 and NSW3-35 (Figure 1). VPRI43306 is a *Citrullus lanatus* (watermelon) leaf and fruit sample collected from Northern Territory properties in 2014 when the virus was first detected in Australia. This isolate was used for the leaf tissue dilution series panel. Isolate NSW3-35, *C. lanatus* leaf material collected during surveillance activities in 2020, was used for contamination of seed for the seed dilution panel due to an insufficient amount of isolate VPRI43306 being available for this process, The two isolates share 99.6% nucleotide identity. Amplicons generated in the single-plex RT-PCRs were directly sequenced using Sanger sequencing. Consensus sequences were assembled to generate the expected 6377 nt portion of the CGMMV genome for both isolates. 

### 2.3. Primer Scheme Analysis

In silico analysis of the 13 primer sets against 137 publicly available CGMMV genomes indicates that the assay designed in this study are likely to amplify most isolates (Appendix A). Mapping of primer CGMMV_600_1_LEFT indicated that 46 of 137 (34%) accessions did not match the 3′ terminal base A, with either a T (37) or G (9) target base present. Primer CGMMV_600_5_RIGHT also showed a mismatch at the 3′ terminal primer base G and target base A for 42 of 137 (31%) sequences. Broadly, European isolates showed a greater number of mismatches across most primer binding regions compared to isolates from Asia, Australia, the Middle East, and North America. 

### 2.4. Tiled Amplicon Sequencing

Tiled amplicon libraries for the plant and seed panels were sequenced for 72 h on the Oxford Nanopore MinION and generated 4,366,037 and 1,583,261 reads that passed quality filtering (Q-score ≥ 7), respectively (Appendix A). Reads less than 300 nt and more than 900 nt were removed, and a total of 4,353,870 and 1,571,746 remaining reads were mapped to the CGMMV reference sequences VPRI43306 and NSW3-35, respectively (Table 1 and Table 2). All plant panel samples covered 100% of reference bases, and seed panel reference base coverage range was 95–100% (Table 1 and Table 2). Read depth coverage of ≥100× was generated across all dilutions of the plant panel and seed panel, except for the 1 in 1000 seed dilution samples (Figure 2 and Figure 3) where coverage dropped below 10 × coverage at NSW3-35 genome positions 2635–2967 and 4640–4955 (amplicons CGMMV600-6 and CGMMV600-10), but >10 × coverage was observed in other regions. Variable coverage was also observed at the 5′ and 3′ ends of the genomes for these samples. 

Tiled amplicon libraries for the four seed interception isolates sequenced on the MinION generated 123,232 raw reads. Following quality trimming and length filtering, a total of 96,483 reads mapped to CGMMV reference genome (GenBank accession KY115174.1) (Table 3). The *C. lanatus* and *C. melo* isolate reads covered 100% of the reference genome with an average depth of 797–2646 reads. The *C. sativus* isolate reads covered 78.96% of the reference genome with an average depth of 5 reads. 

Libraries prepared in duplicate for the KGMMV and ZGMMV isolates, including the multiplex non-template-control products, generated 2001 reads in total, of which only three mapped to the 186K protein CDS region of the reference genome (GenBank accession KY115174.1) (Table 4). 

### 2.5. Metagenomic Sequencing

Metagenomic sequencing generated a total of 387,668,004 and 236,658,289 reads for all plant or seed samples, respectively, that passed quality filtering (Q-score > 20) (Table 1 and Table 2, Appendix A). Reads were mapped to the 6377 nt reference CGMMV genome region used for mapping in the targeted amplicon sequencing, and a total of 2,194,300 plant panel and 8,289,746 seed panel reads mapped. Coverage of reference bases ranged from 0 to 100% for both seed and plant panels. Read depth coverage was ≥10× for the undiluted plant samples and the 10^−1^ and 10^−2^ dilutions but the remainder of dilutions only generated fragmented genome coverage of the 6377 nt reference CGMMV sequence (Figure 4). The single CGMMV contaminated seed and one CGMMV contaminated seed in 10 seed duplicate samples generated ≥10× depth coverage. The one CGMMV contaminated seed in 100 duplicates generated varied depth and genome coverage outputs, and lower dilutions, from 1 in 250 seed to 1 in 1000 seed fell below the ≥10 × coverage level with fragmented coverage (Figure 5).

The seed interception samples generated 30,523,722 reads that passed quality filtering (Q-score > 20) (Table 3). A total of 7766 reads mapped to the CGMMV reference genome KY115174.1. The two *C. lanatus* isolates produced genome coverage of over 99%, with 85–90 times average read depth. The *C. melo* isolate reads covered 80% of the reference sequence with an average depth of 5 reads. The *C. sativus* isolate reads covered 11% of the reference sequence with average depth of less than 1 read.

### 2.6. Sequence Identity Analysis

Consensus sequences were generated for tiled amplicon sequences and where possible, for metagenomic sequences and was based on reference mapping to VPRI43306 and NSW3-35 for plant and seed panel samples, respectively. Tiled amplicon reads generated plant consensus sequences with 99.95–100% nucleotide similarity to the 6377 nt VPRI43306 trimmed sequence. Seed consensus sequences shared 99.70–99.88% nucleotide similarity to the NSW3-35 CGMMV trimmed sequence.

## 3. Discussion

A sensitive and specific TWG-Seq method using a tiled amplicon scheme was successfully developed for the accurate detection and genome assembly of CGMMV from seed and leaf plant samples. The method, which uses PCR-tiling of short (600 nt) amplicons combined with MinION sequencing, was shown to be significantly more sensitive than metagenomic sequencing and at least as sensitive RT-qPCR. TWG-Seq when compared to metagenomic sequencing, improved detection and assembled genomes with reduced fragmentation when applied to low quality archive seed RNA. Specificity of the assay was demonstrated when it was applied to isolates of KGMMV and ZGMMV, *Tobamovirus* species that are closely related to CGMMV, with no significant amplification produced from the multiplex RT-PCRs or nanopore sequencing. No off-target detection of other organisms, including host, by TWG-Seq was observed. Sensitive, informative, and accurate target identification by TWG-Seq will lead to improved diagnostic outcomes when monitoring transmission pathways, including import and export testing of seed and fruit where CGMMV is regulated.

Based on viral load quantification values, the TWG-Seq method for CGMMV detection was at least as sensitive as RT-qPCR but is more informative and accurate since the detection is confirmed by genomic sequence of the target. It can overcome the conundrum faced by diagnosticians associated with false negative results called as a result of high Cq values above RT-qPCR cutoffs, which could be due to target or off target detection or low virus titre. The level of detection (LOD) for the RT-qPCR assay used to detect CGMMV in this study was 150 virus copies per µL (10^−7^, Cq 33.6), yet near complete genomes were assembled for dilutions 10^−8^ to 10^−11^ (copy number < 125), indicating the sensitivity of the methodology. Both RT-qPCR and TWG-Seq were able to detect CGMMV in a one naturally infected seed in 1000 seed dilution and a nearly complete coding region assembled by TWG-Seq. The detection limits of both methods were not reached, and future work could investigate the possibility of detecting one infected CGMMV seed in several 1000 seed and even if only partial genome information is acquired, TWG-Seq would provide greater confidence in detection compared to RT-qPCR. 

TWG-Seq is similar to a targeted genomic sequencing (TG-Seq) approach that was recently described and based on a direct enrichment method for cDNA of plant viruses in which specific genome regions of four plant viruses were amplified using multiplex PCR and sequenced using the Illumina MiSeq platform [30]. TG-seq was found to be highly sensitive when compared to conventional multiplex PCR analysed by gel electrophoresis, and plant host reads made up less than 1% of total reads. Likewise, the elimination of host reads, and enrichment of virus reads in the CGMMV TWG-Seq assay evaluated here, resulted in high sensitivity when compared to a metagenomic approach. Reference base coverage of 100% was produced across all dilution samples with the exception of the lowest concentration seed dilutions. At 95% coverage, this is sufficient to confirm the presence of virus. 

Using TWG-Seq, average depth coverage of ≥100 × was achieved for all CGMMV seed and plant dilutions, however when examined across the full genome, there were regions of reduced coverage for low concentration seed samples. Reduced coverage was also observed for the oldest of the archived RNA seed samples, *Cucumis sativus*_2014-1. Uniform amplification of each tile region can be achieved with a focus primer design and ensuring all possible strains are included in the design process. The incorporation of additional primers for low coverage or missing regions [31] can also accommodate future mutations [24], whilst the introduction of degenerate primers has been shown to improve assay sensitivity and coverage of evolving strains if full genomes are required for diversity studies [32,33]. Increasing the targeted amplicon length [34,35] can improve coverage, but is effected by RNA quality and quantity. The provision of alternative assays to accommodate these conditions would be recommended for diagnostic usage.

As stated previously TWG-Seq generates sequence that confirms a detection and thus reduces the risk of false positive that can be generated by other methods because of off target amplification, such as those generated by RT-qPCR results in the leaf panel buffer (zucchini leaf homogenate. TWG-Seq did not generate sequence of other organisms associated with off-target detection observed in uninfected tissues. Australian CGMMV isolates were effectively detected using the TWG-Seq primer scheme, whilst closely related viruses KGMMV and ZGMMV did not produce significant reads, providing further support for the specificity of this assay. 

Initial primer design incorporated isolates from Asia, Australia, the Middle East, and North America. Subsequent in silico analysis of the 13 primer sets against 137 publicly available CGMMV genomes revealed a high number of mismatches throughout the primer binding regions for the majority of primers when aligned with European isolates. Two primers in particular showed mismatches at the 3′ terminal base. Amplification efficiency has been shown to be influenced by this type of mismatch, with a G:A or A:G variance resulting in a 100-fold efficiency reduction [36]. However, efficiency is not significantly impacted when a T is located at the 3′ end of the primer or target. Multiple mismatches at the 3′ end of the primer can also influence the efficacy of the assay, with additional differences 1, 2 or 3 bases from a non-T 3′ terminal nucleotide reported to significantly reduce yield. These effects will impact the assembly of complete genomes, however enough sequence should be generated by the matching primers from these CGMMV isolates for diagnostic purposes, although this needs to be confirmed by testing TWG-Seq against these strains. 

TWG-Seq is sensitive to contamination from aerosolised PCR products and flow cell carryover [11,25,26,35]. The overamplification of high concentration samples increases the risk of amplicon contamination of low virus copy, especially when sequentially processing a dilution series. This could explain the presence of CGMMV reads in the seed buffer control sample. Individual flow cells were used for the sequencing of plant and seed dilutions series in this study. Contamination problems can be managed by physical separation of pre-PCR spaces, attention to integrity of reagents and cleanliness of equipment and work benches [25]. The use of controls such as no-template water controls at the cDNA and PCR steps included in the sequencing run can provide an indication of contamination, and bioinformatic analysis of negative sample reads to positive sample reads can signify level of contamination present, which can be taken into consideration during analysis and help to set a cut-off for defining a positive TWG-Seq result.

The potential for the combination of multiple targets should also be investigated. In addition to CGMMV, current import conditions for cucurbitaceous vegetable seed species imported for sowing also requires mandatory ELISA testing for KGMMV, ZGMMV and melon necrotic spot virus (MNSV) for host seed species of these pathogens (https://www.agriculture.gov.au/biosecurity-trade/import/industry-advice/2020/80-2020 (accessed on 28 July 2022)). Combining targets for tiled amplicon sequencing of these tobamoviruses would be highly beneficial but would require extensive optimization and validation to establish a high level of confidence. Combining targets within the one assay has the potential to be used for surveillance and diagnostics, delivering fast identification to inform management options and improve monitoring of transmission pathways, including imported plant products.

An optimised approach for the detection of CGMMV using tiled amplicon sequencing has potential to be implemented in a diagnostics setting, providing rapid and reliable confirmation of PCR and ELISA results, especially those near current cut-off values. The MinION is a portable and affordable sequencing option for laboratories with limited HTS resources and is a viable alternative to Sanger sequencing. Sensitivity of tiled amplicon assays means the confident screening of larger pools of plant tissues for surveillance could also be accomplished. This targeted tool could also be used routinely during seed production and by nurseries and growers for early virus detection so that they can enact management strategies to prevent further spread. 

## 4. Materials and Methods

### 4.1. Plant Materials 

CGMMV positive leaf and fruit samples were obtained during cucurbit crop surveys carried out in July 2014 on NT properties where severe mosaic and stunted foliar development was observed on *Citrullus lanatus* (watermelon) plants. Leaf and fruit samples were submitted to Crop Health Services (CHS), Agriculture Victoria, by the Northern Territory Department of Industry, Tourism and Trade, tested positive for CGMMV by RT-PCR (data not shown). Isolate VPRI43306 is maintained as freeze-dried material. In 2020, symptomatic *C. lanatus* leaves were collected during surveillance activities in New South Wales and tested positive for CGMMV by RT-PCR (data not shown) and high through-put sequencing (unpublished). This isolate, NSW3-35 is maintained as freeze-dried material. Uninfected *Cucurbita pepo* (zucchini) leaf tissue was sourced from a home garden (author) in Victoria, Australia with no history of CGMMV. Seed naturally contaminated with CGMMV, including one lot each of *Cucurbita maxima* (pumpkin) and *C. maxima X C, moschata* hybrid seed and two lots of *Cucurbita moschata* (pumpkin) seed, were kindly provided by Aviv Dombrovsky (The Volcani Centre, Rishon LeZion, Israel). Uninfected melon seed was sourced from in-house stock that had previously tested negative for CGMMV by ELISA. KGMMV (Part Number LPC 65001) and ZGMMV (Control No. Adg/080716/20) positive controls were obtained from Agdia, Inc. (TASAG ELISA and Pathogen Testing Service, New Town, TAS, Australia) and LOEWE Biochemica GmbH (Sauerlach, Germany), respectively. Seed interception isolate *Cucumis melo*_2015-1 was provided by Elizabeth Macarthur Agricultural Institute (EMAI), NSW Department of Primary Industries (NSWDPI) as extracted RNA. Seed interception isolates *Cucumis sativus*_2014-1, *Citrullus lanatus*_2018-1 and *Citrullus lanatus*_2018-2 were provided by CHS, Agriculture Victoria as extracted RNA.

### 4.2. Preparation of Virus Panels

A panel of 13 RNA extracts was prepared using freeze-dried material of CGMMV isolate VPRI43306. The CGMMV homogenate was prepared using 0.5 g tissue in 10 mL lysis buffer [37] and the uninfected tissue homogenate used 3 g fresh uninfected zucchini leaf tissue in 30 mL lysis buffer [37]. A tenfold dilution series ranging from 10^−1^ to 10^−8^ of the CGMMV homogenate in uninfected tissue homogenate was prepared. A single RNA extraction was carried out on each dilution, the undiluted CGMMV homogenate and the uninfected zucchini homogenate and the KGMMV and ZGMMV isolates, using the RNeasy Plant Mini kit (QIAGEN, Doncaster, Vic, Australia) following the manufacturer’s instructions. The dilution series is referred to as the “plant panel”. 

Artificially contaminated cucurbit seeds were used to generate a seed pool series from which RNA was extracted. To calculate the average titre at which to contaminate seeds, the CGMMV titre in individual naturally infected pumpkin seeds and *C. maxima* x *C. moschata* hybrid seeds was estimated. Five seeds from four infected seed lots were used for RNA extraction. Individual seeds were ground in 1 mL of ELISA extraction buffer containing phosphate-buffered saline (PBS) with 0.0025% Tween^®^ 20 and 2% polyvinylpyrrolidone 40,000. For RNA extraction, 100 µL of each seed homogenate was used for RNA extraction using the RNeasy Plant Mini Kit (Qiagen, Doncaster, VIC, Australia) according to manufacturer instructions. Triplicates of each individual seed RNA extract were used for CGMMV RT-qPCR [29] (described in Section 4.3) alongside a ten-fold eight-point dilution series of transcribed CGMMV RNA standards (described in Section 4.4). The actual concentration of CGMMV in each infected seed was then calculated by comparing their Cq values to the transcribed RNA standards of known molecule copy number.

To prepare artificially contaminated seeds, one gram of freeze-dried CGMMV isolate NSW3-35 was homogenised in 5 mL, 10 mL, and 15 mL of homogenization buffer (55× phosphate-buffered saline, 0.25% (vol/vol) Tween^®^ 20 and 2% (wt/vol) PVP-40). Three healthy virus-free *Cucumis sativus* (cucumber) seeds were added to each of the CGMMV homogenates and soaked for 2 min in an orbital shaker. Seeds were dried on paper towel for 30 min. Uninfected seeds were also soaked in homogenization buffer for use as a negative control. Total RNA was extracted from each contaminated seed and used for CGMMV RT-qPCR described below (Section 4.3). An average Cq value of 21 was observed for the seeds soaked in the 5 mL virus homogenate. With a Cq value of 20 observed for the naturally infected seed, the 5 mL homogenate dilution was selected for further seed contamination.

Duplicate seed pools were created that contained one contaminated seed in a total of 10, 100, 250, 500 and 1000 seeds, and then crushed and homogenized in the homogenisation buffer, which was added at the rate of 5 mL for each 1 g of seed. Duplicate contaminated single seeds were also used for RNA extraction and testing. RNA was extracted in triplicate from each seed pool and the uninfected control seed samples. RNA was also extracted in triplicate from the homogenized plant tissue used to soak seeds as a positive control and the homogenisation buffer, which was used as an extraction control. For each triplicate, 100 µL of the homogenate from each sample and the homogenization buffer control was mixed with 450 µL of RLT buffer provided with the RNeasy Plant Mini Kit (QIAGEN, Doncaster, VIC, Australia) and RNA was then extracted according to the manufacturer’s instructions. A total of 45 RNA extracts were generated and are referred to as the “seed panel”.

### 4.3. RT-qPCR Conditions 

Quantitative RT-PCR (RT-qPCR) [29] on a QuantStudio 3 thermocycler (Thermo Fisher Scientific, Scoresby, VIC, Australia) was used for detection of CGMMV in standards and seed and plant panels. A GoTaq^®®^ Probe 1-step RT-qPCR System (Promega Corporation, Alexandria, NSW, Australia) was used and the final reaction volume of 20 µL contained 1 µL of each 10 µM primer, 0.5 µL of 10 µM probe, 10 µL GoTaq^®^ Probe qPCR Master Mix with dUTP, 0.4 µL GoScript™ RT Mix for 1-Step RT-qPCR and 3 µL RNA template. Three technical replicates were analysed for every seed and plant panel sample, the transcribed RNA standards used for absolute quantification CGMMV in seed and plant samples and the no-template control.

### 4.4. Transcribed RNA Standards for Absolute Quantification of CGMMV by RT-qPCR

The transcribed RNA standards for absolute CGMMV quantification were produced as described previously with some modifications [38]. Briefly, a synthetic CGMMV fragment (538 nt) containing binding sites from primers and probes for the RT-qPCR [29] and a T7 promoter for RNA transcription was designed and procured (Thermo Fisher Scientific, Scoresby, VIC, Australia) (Appendix A). In vitro RNA transcription of the synthetic CGMMV fragment was done using a MEGAScript^®®^T7 Kit (Thermo Fisher Scientific, Scoresby, VIC, Australia) then DNA was removed from the transcribed RNA using Turbo DNase (Thermo Fisher Scientific, Scoresby, VIC, Australia); both kits were used according to the manufacturer’s instructions. After DNase treatment the CGMMV RT-qPCR primers [29] were used in combination with a Power SYBR Green Quantitative PCR System (Thermo Fisher Scientific, Scoresby, VIC, Australia) to detect any remaining DNA by qPCR on a QuantStudio3 Thermocycler, and repeated until no DNA could be detected. The concentration of transcribed RNA was measured using a NanoDrop™ 1000 Spectrophotometer (Thermo Fisher Scientific, Scoresby, VIC, Australia) and converted to molecule copy number using an online calculator (http://endmemo.com/bio/dnacopynum.php (accessed on 2 March 2022)) with the ssRNA option selected. A tenfold serial dilution of the transcribed RNA in water was used to create a standard curve from 3 ng/µL (60,204,394,921 copies) to 0.3 fg/µL (6020 copies) and was used to quantify CGMMV in samples (Appendix A; Appendix A).

### 4.5. Multiplex PCR Primer Design

The web-based primer design tool, Primal Scheme (https://primalscheme.com (accessed on 28 April 2021)) [25] was used to design multiplex primer sets for the amplification of the CGMMV genome. Briefly, full CGMMV genomes of approximately the same length were aligned using Clustal Omega [39]. Sequences with maximum sequence divergence of 5% or more were removed from the alignment, as were sequences with 99–100% identity to other genomes (Appendix A). The final alignment was submitted to the Primal Scheme portal with the desired PCR amplicon length set at 600 bp with neighboring amplicon overlap of 100 nt.

Each of the primer sets were used in a single-plex PCR reaction to determine if they could amplify CGMMV in the two isolates selected for the panel preparation, the zucchini dilution homogenate, and the KGMMV and ZGMMV isolates. Using 7 µL total RNA, reverse transcription was done using ProtoScript II First Strand cDNA Synthesis Kit (New England Biolabs, Notting Hill, VIC, Australia) and Random Primer Mix (New England Biolabs, Notting Hill, VIC,, Australia) according to the manufacturer’s standard protocol. PCR reactions contained 1 µL of cDNA reaction as input and were set up using Q5 High-Fidelity DNA Polymerase (New England Biolabs, Notting Hill, VIC, Australia) using the manufacturer’s recommended protocol. The amplicons from each of the 13 PCR reactions were visualised by electrophoresis in 1.5% agarose gels stained with SYBR™ Safe DNA gel stain (Thermo Fisher Scientific, Scoresby, VIC, Australia) and submitted to the Australian Genome Research Facility (AGRF, Melbourne, VIC, Australia) for direct Sanger sequencing in both directions. Assembly of forward and reverse sequences was carried out using Geneious Prime (version 2022.0.1) (Biomatters Ltd.) (Figure 6).

To determine if the designed primer scheme would effectively amplify full genomes of all known CGMMV isolates, in silico analysis was performed. Using Geneious Prime (version 2022.0.1) (Biomatters Ltd.), primers were mapped to an alignment of 137 publicly available CGMMV genomes, with up to 6 binding region mismatches permitted.

### 4.6. Multiplex Tiling PCR

Primer pools for the multiplex tiling PCR assays were prepared by combining an equal volume of each 100 µM primer stock for both forward and reverse primers for alternate regions (Appendix A). These were diluted to 10 µM working concentration as recommended by Quick et.al. [25]. Using 2.5 µL of cDNA prepared according to method for single-plex PCRs, a multiplex PCR reaction was carried out for each primer pool with 0.25 µL Q5 High-Fidelity DNA Polymerase (New England Biolabs, Notting Hill, VIC, Australia), 5 µL Q5 reaction buffer (5×), 0.5 µL 10 mM dNTPs, primer pool (1 or 2) with 0.015 µM final concentration and RNase-free water to a final volume of 25 µL. Cycling conditions used: 98 °C for 30 s, and 39 cycles of 98 °C for 15 s, 65 °C for 5 min. For each sample, the PCR products from both pools were combined and purified using AMPure XP beads (Beckman Coulter, Mount Waverley, VIC, Australia) and quantified using the Qubit™ Flex Fluorometer and Qubit™ 1X dsDNA HS Assay Kit (Thermo Fisher Scientific, Scoresby, VIC, Australia). Amplicons were normalised to achieve a total of ~0.3 pM per flow cell [25] by dividing the total library input of 120 ng by the number of barcodes being used.

### 4.7. Tiled Amplicon Sequencing and Bioinformatics

Based on a final library concentration of ~0.3 pM per flow cell, a total input of 120 ng of the final pool of 600 bp amplicons, generated by the Pool 1 and Pool 2 multiplex PCR assays, was used for library preparation. End-repair and dA-tailing were performed using the NEB Next Ultra II End-repair/dA-tailing Module (New England Biolabs, Notting Hill, VIC, Australia). Normalised amplicons (20 µL) were combined with 2.8 µL NEBNext Ultra II End Prep Reaction Buffer and 1.2 µL NEBNext Ultra II End Prep Enzyme Mix, and incubated for 5 min at 20 °C, followed by 5 min at 65 °C. AMPure XP bead clean-up was performed using 0.8 × reaction volume, eluting in 10 µL EB buffer. Barcode ligation using the Native Barcoding Kit (Oxford Nanopore Technologies, Oxford, UK) and NEB Blunt/TA Ligase Master Mix (NEB, cat. no. M0367) was performed using 2.5 µL barcode per sample (combined PCR amplicons from pool 1 and pool 2) and 12.5 µL NEB Blunt/TA Ligase Master Mix (NEB, cat. no. M0367). Incubation of barcode-ligated amplicons at room temperature (20 °C) for 10 min was followed by 65 °C for 10 min. All barcode ligation reactions were pooled and purified using 0.8× AMPure XP beads and eluted in 30 µL RNase-free water. Adapter ligation was carried out using the Nanopore Ligation Sequencing Kit (Oxford Nanopore Technologies, Oxford, UK) according to the ONT Native Barcoding Kit 96 with amplicons protocol (available from https://community.nanoporetech.com/docs/prepare/library_prep_protocols (accessed on 24 June 2021)). Libraries were sequenced on the MinION (Oxford Nanopore Technologies, Oxford, UK) using R.9.4 flow-cell and MinKNOW software v.4.3.25 (Oxford Nanopore Technologies, Oxford, UK).

Reads were base-called and demultiplexed using Guppy (Version 5.0.16) and trimmed using Porechop (version 0.2.4). Filtering to remove chimeric reads was performed using ARTIC filter (version 1.2.1) with minimum read length cut-off set at 300 nt and maximum length cut-off set at 900 nt. Reads were aligned to the trimmed CGMMV contig generated using Illumina metagenomic sequencing (Section 4.8) using BBMap (Version 38.87) [40] with default settings. SAMtools (version1.10) was used to sort and index reads and calculate read depth. Primer sequences were soft-clipped using BamCLIPPER (v1.0.0) [41] (options -u 50 -d 50). Draft consensus sequences were called from alignments using BCFtools (version 1.12) mpileup and call (options -Q 7 -d 1000). Medaka mini-align (version 1.4.3) was used to map base-called reads to draft consensus sequences. Variants were filtered to remove calls with depth less than 20, a quality score less than 50 and indel support fraction less than 70% and applied to draft consensus sequences to generate final consensus sequences. 

### 4.8. Metagenomic Sequencing and Bioinformatics

Individual libraries were prepared for the CGMMV infected tissues used to prepare seed and plant panels and each seed panel sample including buffer controls. Triplicate libraries were prepared for each plant panel sample including the zucchini dilution homogenate RNA. Libraries were prepared using the NEBNext^®®^ Ultra™ II RNA Library Prep Kit for Illumina^®®^ according to the manufacturer’s instructions. The Qubit™2.0 Fluorometer and a Qubit dsDNA HS assay kit (Thermo Fischer Scientific, Scoresby, VIC, Australia) were used to quantify libraries and the 2200 TapeStation (Agilent Technologies) and HSD1000 ScreenTape assays (Agilent Technologies) used determine library sizes. The final pooled library was sequenced on an Illumina NovaSeq 6000 instrument (2 × 150 bp reads).

Demultiplexed sequence reads were quality filtered (Quality score > 20, minimum read length of 50), adapter sequences trimmed and read pairs validated using Trim Galore! (version 0.6.4) [42]. Read pairs were merged using fastp (version 0.20.0) [43]. 

De novo assembly of the undiluted plant homogenates used to prepare the leaf and seed panels was performed with SPAdes (version 3.15.2) [44] using options --rnaviral and -k 127,107,87,67,31. Assembled contigs of 1000 nt or more were analysed using BLASTn (version 2.9.0) [45] and CGMMV contigs of 6514 nt and 6628 nt for VPRI43306 and NSW3-35, respectively, were selected for mapping of reads generated by metagenomic sequencing for all other seed and plant panel samples and the uninfected or buffer controls. Tiled amplicon primers were mapped to each contig, and the sequences trimmed to the tiled amplicon target length of 6377 nt bounded by the most distal primers located at the 5′ and 3′ regions of the genome to enable an accurate comparison of sequence outputs of the metagenomic and tiled amplicon sequencing. Metagenomic sequence reads were mapped to 6377 nt sequences and SAMtools (version1.10) was used to calculate read depth. Consensus sequences were called from alignments using BCFtools (version 1.12) and required a minimum of 1× coverage depth and a minimum support fraction of 20% for calling ambiguities.

### 4.9. Sequence Accuracy Assessment

The nucleotide similarity of tiled amplicon and metagenomic generated CGMMV genome consensus sequences was assessed by pairwise alignment using MAFFT (Version 1.4.0) in Geneious Prime (Version 2022.1.1).

## Figures and Tables

**Figure 1 plants-11-02716-f001:**
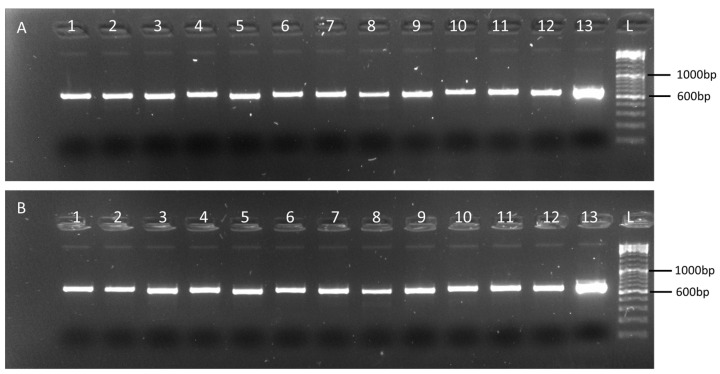
The 600 bp single-plex PCR products for each of the thirteen primers pairs (600-1–600-13, lanes 1 to 13) used to generate overlapping amplicons for tiled amplicon sequencing for (**A**) VPRI43306 and (**B**) NSW3-35. The amplicons were visualised on 1.5% agarose gel with SYBR™ Safe DNA gel stain (Thermo Fisher Scientific, Scoresby, Australia). A 100 bp DNA ladder (Thermo Fisher Scientific Scoresby, Australia) was used for product size estimation.

**Figure 2 plants-11-02716-f002:**
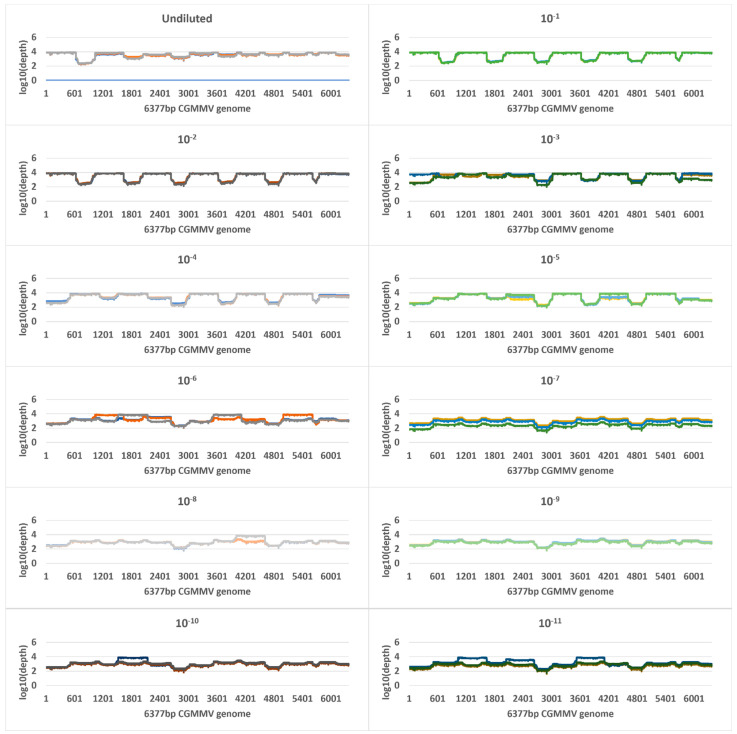
Genome coverage and read depth profiles of plant panel tenfold dilution series samples, from undiluted to 1 in 100,000,000,000 (10^−11)^ across the 6377 nt cucumber green mottle mosaic virus (CGMMV) genome (VPRI43306) detected using tiled amplicon nanopore sequencing. Each dilution was tested in triplicate and the results for each technical replicate are shown as different colours. Depth is in standard logarithmic scale.

**Figure 3 plants-11-02716-f003:**
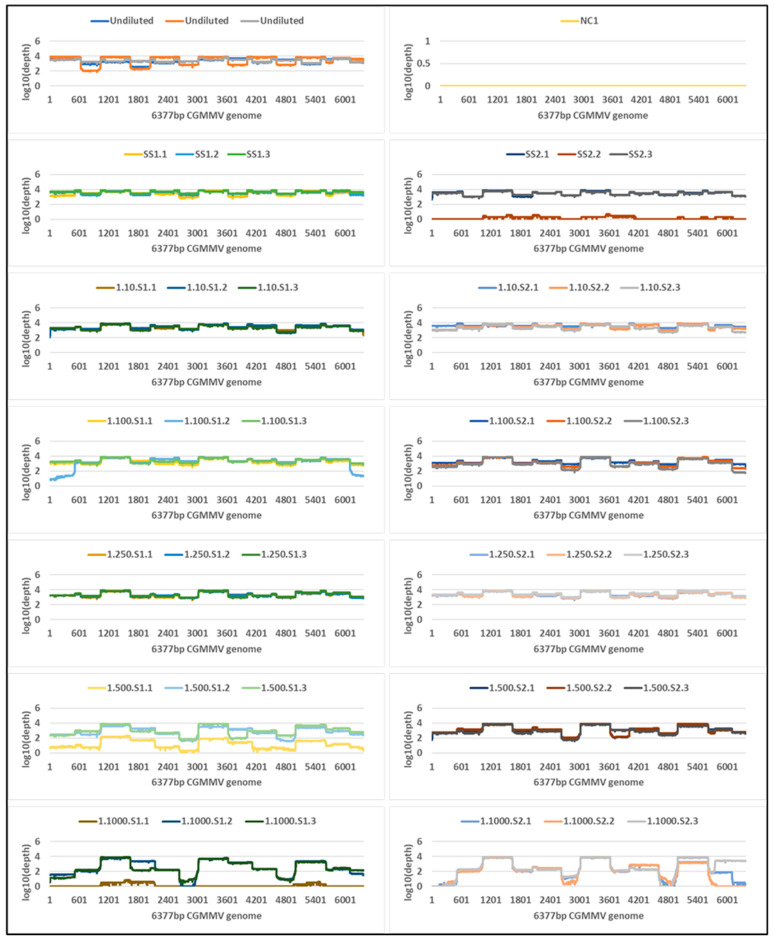
Genome coverage and read depth profiles of seed panel dilution series samples, from undiluted to 1 seed in 1000 across the 6377 nt reference cucumber green mottle mosaic virus (CGMMV) genome (NSW3-35) detected using tiled amplicon nanopore sequencing. Biological replicates are shown in each profile. Each biological replicate was tested in triplicate and the results for each technical replicate are shown as different colours. Depth is in standard logarithmic scale.

**Figure 4 plants-11-02716-f004:**
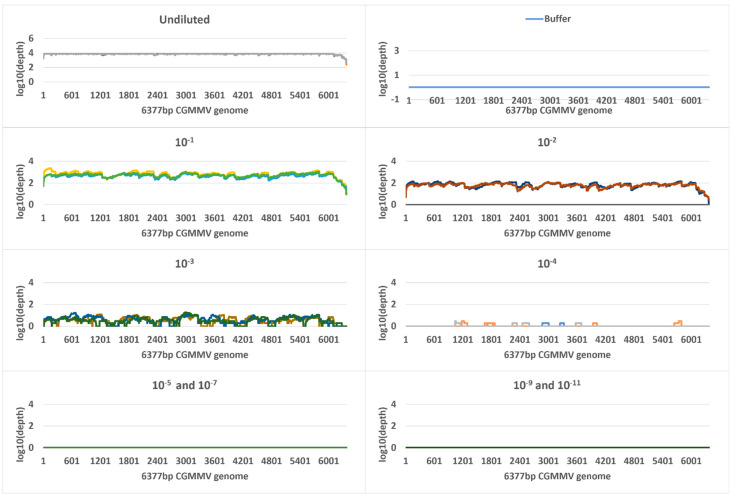
Genome coverage and read depth profiles of plant panel tenfold dilution series samples, from undiluted to 1 in 100,000,000,000 (10^−11^) across the 6377 nt cucumber green mottle mosaic virus (CGMMV) genome (VPRI43306) detected using Illumina metagenomic sequencing. Each plant panel dilution was tested in triplicate and the technical replicates are shown as different colours. Depth is in standard Logarithmic Scale.

**Figure 5 plants-11-02716-f005:**
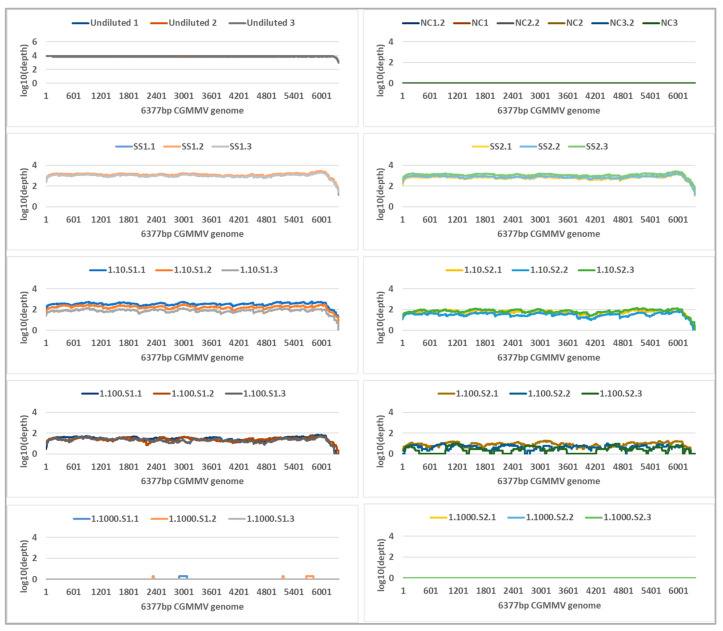
Genome coverage and read depth profiles of seed panel dilution series samples, from undiluted to 1 in 1000 across the 6377 nt reference cucumber green mottle mosaic virus (CGMMV) genome (NSW3-35) detected using Illumina metagenomic sequencing. Biological replicates are shown in each profile. Each biological replicate was tested in triplicate and the results for each technical replicate are shown as different colours. Depth is in standard logarithmic scale.

**Figure 6 plants-11-02716-f006:**
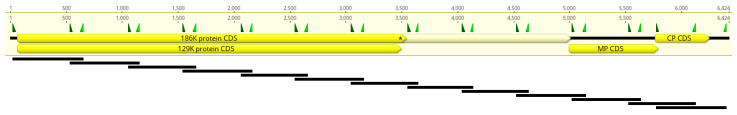
Diagram showing alignment of PCR amplicons (black lines) to the cucumber green mottle mosaic reference genome (GenBank Accessions No. NC_001801.1). Coding sequence regions for the 186K protein, 129K protein, movement protein and coat protein are shown. Multiplex primers are shown at predicted binding sites, left (forward) primers are dark green and right (reverse) primers are in light green. (Geneious version 2022.0 created by Biomatters. Available from https://www.geneious.com).

**Table 1 plants-11-02716-t001:** The cucumber green mottle mosaic virus (CGMMV) plant panel dilution series samples and buffer control that were tested by quantitative RT-PCR, tiled amplicon nanopore sequencing and the untargeted metagenomic short read sequencing methods. Absolute quantification of virus copy number per µL of RNA extract was calculated from the quantitative RT-PCR cycle quantification (Cq) value when compared to a standard curve (Appendix A). Tiled amplicon and the untargeted metagenomic sequence data is presented for the number (No.) of sequence reads generated for each sample after quality and length trimming, number of reads mapping to the 6377 nt reference CGMMV sequence (isolate VPRI43306), percentage (%) of CGMMV reference sequence bases covered, and average number of reads at each reference base (Average read depth). Values represent the average of three technical replicates. Standard deviation (SD) is given in brackets.

Sample	Cq	Virus Copies Per µL of RNA (SD)	Tiled Amplicon—MinION				Untargeted Metagenomic—NovaSeq			
			Reads Used	Reads Mapped to CGMMV (SD)	% Ref. Bases Covered	Average Read Depth	Reads Used	Reads Mapped to CGMMV (SD)	% Ref. Bases Covered	Average Read Depth
Undiluted ^a^	9.5	724,858,724 (505,575,547)	119,982	119,737 (19,326)	100	4633	7,948,784	705,257 (99,781)	100	7535
10^−1^	11.8	148,495,975 (48,814,341)	428,078	427,838 (39,232)	100	5413	8,565,097	24,079 (5912)	100	585
10^−2^	15	18,581,696 (4,560,628)	285,974	285,786 (44,622)	100	5331	10,941,741	2842 (1648)	100	69
10^−3^	18	4,400,508 (5,288,129)	177,081	176,916 (107,377)	100	4432	10,444,239	185 (19)	95	5
10^−4^	21.5	322,122 (174,102)	220,367	220,147 (45,775)	100	4301	10,212,900	15 (4)	25	0
10^−5^	25.7	20,532 (3610)	85,189	85,031 (29,087)	100	3293	9,821,045	0	0	0
10^−6^	29	2316 (250)	60,563	60,300 (14,479)	100	2251	12,239,795	0	0	0
10^−7^	33.6	125 (26)	9794	9433 (6603)	100	893	15,300,104	0	0	0
10^−8^	37.1	15 (10)	10,721	18,509 (15,356)	100	1098	10,047,613	3	6.6	0.07
10^−9^	37.9	23	11,992	11,225 (1771)	100	1060	7,751,478	0	0	0
10^−10^	37.3	17 (17)	26,385	17,084 (10,639)	100	1201	14,537,532	0	0	0
10^−11^	38	8 (4)	8725	18,494 (18,593)	100	1293	11,493,244	0	0	0
Buffer ^b^	36.8	22 (21)	1532	0	0	0	10,699,014	1	2.4	0.02
PCR NTC ^c^	NA	NA	7	0	0	0	NA	NA	NA	NA

^a^ Undiluted CGMMV leaf homogenate; ^b^ Zucchini leaf homogenate; ^c^ Pooled multiplex PCR non-template control.

**Table 2 plants-11-02716-t002:** The cucumber green mottle mosaic virus (CGMMV) seed panel dilution series samples and buffer control that were tested by quantitative RT-PCR, tiled amplicon nanopore sequencing and the untargeted metagenomic short read sequencing methods. Absolute quantification of virus copy number per µL of RNA extract was calculated from the quantitative RT-PCR cycle quantification (Cq) value when compared to a standard curve (Appendix A). Tiled amplicon and the untargeted metagenomic sequence data is presented for the number (No.) of sequence reads generated for each sample after quality and length trimming, number of reads mapping to the 6377 nt reference CGMMV sequence (isolate NSW3-35), percentage (%) of CGMMV reference sequence bases covered, and average number of reads at each reference base (Average read depth). Values represent the average of three technical replicates. Standard deviation (SD) is given in brackets.

Sample	Cq	Virus Copies Per µL of RNA (SD)	Tiled Amplicon—MinION				Untargeted Metagenomic—NovaSeq			
			Reads Used	Reads Mapped to CGMMV (SD)	% Ref Bases Covered	Average Read Depth	Reads Used	Reads Mapped to CGMMV (SD)	% Ref Bases Covered	Average Read Depth
Undiluted ^a^	8.4	1,526,567,296 (1,113,748,195)	48,837	48,827 (34,308)	100	3417	2,844,308	2,705,523 (246,410)	100	7820
Single seed 1	13.8	44,181,975 (3,663,819)	61,585	61,543 (13,126)	100	4284	6,586,247	28,232 (4263)	100	1149
Single seed 2	13.8	35,809,461 (3,057,510)	26,005	25,996 (22,799)	100	3445	4,967,016	21,700 (7948)	100	873
1 seed in 10: Seed 1	18.1	3,016,299 (2,079,368)	34,912	34,901 (4711)	100	3029	5,231,106	4940 (3312)	100	204
1 seed in 10: Seed 2	16.3	9,965,820 (7,389,633)	67,942	67,919 (36,596)	100	3837	8,859,498	1472 (423)	100	57
1 seed in 100: Seed 1	20.9	429,383 (121,406)	35,503	35,483 (9375)	100	2847	9,101,950	684 (124)	100	27
1 seed in 100: Seed 2	19.3	1,190,964 (216,733)	42,308	42,295 (6359)	100	2639	6,293,380	142 (79)	94	6
1 seed in 250: Seed 1	21.4	331,757 (116,464)	36,270	36,259 (4850)	100	2871	5,577,062	298 (60)	99	12
1 seed in 250: Seed 2	20.3	628,026 (195,273)	50,390	50,377 (11,703)	100	3224	4,846,547	209 (146)	96	8
1 seed in 500: Seed 1	24.3	47,418 (6915)	17,362	17,351 (17,552)	100	1939	4,108,499	17 (9)	45	1
1 seed in 500: Seed 2	24.4	46,624 (7676)	50,839	50,823 (20,825)	100	2565	4,298,383	28 (11)	61	1
1 seed in 1000: Seed 1	30.4	947 (106)	12,185	12,177 (10,585)	98	1569	4,164,645	4 (2)	17	0
1 seed in 1000: Seed 2	27.0	8894 (1410)	39,969	39,958 (7833)	95	2031	2,063,940	0	0	0
NC1 ^b^	UD	0	NA	NA	NA	NA	5,672,805	0	0	0
NC2 ^b^	UD	0	NA	NA	NA	NA	4,270,709	0	0	0
PCR NTC ^c^	NA	NA	10,941	11	48	1	NA	NA	NA	NA

^a^ Undiluted CGMMV (NSW3-35) homogenate; ^b^ Homogenisation buffer; ^c^ Pooled multiplex PCR non-template control.

**Table 3 plants-11-02716-t003:** The seed interception samples that were tested by tiled amplicon nanopore sequencing and the untargeted metagenomic short read sequencing methods. Tiled amplicon and the untargeted metagenomic sequence data is presented for the number of sequence reads generated for each sample after quality and length trimming, number of reads mapping to the 6377 nt reference CGMMV sequence (GenBank Accession KY115174.1), percentage (%) of CGMMV reference sequence bases covered, and average number of reads at each reference base (Average read depth).

	Tiled Amplicon—MinION				Untargeted Metagenomic—NovaSeq			
Seed Interception Isolate	Reads Used	Mapped Reads	Percentage of Reference Bases Covered	Average Read Depth	Reads Used	Mapped Reads	Percentage of Reference Bases Covered	Average Read Depth
*Citrullus lanatus*_2018-1	8701	8633	100	797.156	13,808,999	3603	99.16	85.3656
*Citrullus lanatus*_2018-2	10,121	10,012	100	916.348	9,726,986	3858	99.21	90.6358
*Cucumis melo*_2015-1	83,828	77,781	100	2646.63	5,335,679	291	80.49	5.86938
*Cucumis sativus*_2014-1	1003	57	78.96	5.19711	1,652,058	14	11.38	0.265608

**Table 4 plants-11-02716-t004:** Kyuri green mottle mosaic virus (KGMMV) Part Number LPC 65001 and zucchini green mottle mosaic virus (ZGMMV) Control No. Adg/080716/20 positive controls were tested in duplicate by tiled amplicon nanopore sequencing. Data is presented for the number of sequence reads generated for each sample after quality and length trimming, number of reads mapping to the 6377 nt reference CGMMV sequence (GenBank Accession KY115174.1), percentage (%) of CGMMV reference sequence bases covered, and average number of reads at each reference base (Average read depth).

Samples	Reads Used	Mapped Reads	Average Coverage with Deletion	Percentage of Reference Bases Covered
KGMMV-1	316	2	0.105	5.49
KGMMV-2	213	1	0	8.04
ZGMMV-1	1090	0	0	0
ZGMMV-2	132	0	0	0
NTC-1	106	0	0	0
NTC-2	144	0	0	0

## Data Availability

Not applicable.

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
