# Peer review of "Targeted Whole Genome Sequencing (TWG-Seq) of Cucumber Green Mottle Mosaic Virus Using Tiled Amplicon Multiplex PCR and Nanopore Sequencing"

_plants, 2022, doi:10.3390/plants11202716_

Round 1

Reviewer 1 Report

The manuscript “Tiled amplicon detection of cucumber green mottle mosaic virus using multiplex PCR and Nanopore sequencing” describes a method in which CGMMV specific primers were used to amplify, in small overlapping fragments of 600 nt, the full coding sequence of the virus, and then amplicons were prepared for Nanopore-based sequencing using MinION. The protocol starts with random primers to make cDNA of total RNA (in a ten-fold dilution series) isolated from CGMMV infected cucurbit leaves and seeds. Next, in two multiplex PCRs cDNA were used as template to make overlapping amplicons of CGMMV. Virus detection sensitivity in terms of percent genome coverage of CGMMV was compared between Nanopore and Illumina sequencing. Contrary to Nanopore sequencing, in Illumina sequencing total RNA from leaves and seeds were sequenced directly. No CGMMV specific primers and RT-PCR based enrichment procedure was included for Illumina sequencing. Based on results from these experiments, it was claimed that Nanopore sequencing is more sensitive for target detection than Illumina. In my opinion, this is not a fair comparison of two different sequencing methods. If you want to compare detection capabilities of Nanopore sequencing and Illumina sequencing, the best approach is either to provide total RNA as it is or provide target-enriched starting material for library preparation and sequencing. There are several reports where viral targets have been enriched for Illumina sequencing. If you want to present this data, as it is, I’ll suggest not to compare sequencing approaches for their sensitivities. This unfair comparison will not convey readers the right message about the sensitivities of two different sequencing chemistries. I think this approach by itself is good enough for publication. This is just a suggestion that if you want to include Illumina sequencing data, use it to compare sequencing quality of consensus sequences or may be of individual sequence reads.

                There are reports of Nanopore sequencing of target enriched nucleic acids (Leiva et al 2020 Microbiol Resour. Announc.; Chittarath et al 2021 Plant Dis.). I’ll advise to include those reports in your discussions and tell readers how this approach is different/useful from others.

Since this approach is specifically for the detection of CGMMV and uses 26 CGMMV specific primer; to make this manuscript more informative, I would like to raise some questions. What if there are CGMMV isolates with mismatches in one or more than one primer binding regions; how these mismatches are going to affect sequencing; can these primers amplify European, Asian or both variants of CGMMV; what are shortcomings of this approach for viruses with high genetic diversity, recombination, detection of novel viruses etc.; an in silico analysis of these primers and all available isolates of CGMMV will be useful. Can you enhance variant detection by adding some primer degeneracy?

Suggesting some other minor changes:

Line 249                change “more sensitive that RT-qPCR” to “more sensitive than RT-qPCR”

Line 396                change “natural” to “naturally”

Line 397                C. maxima X C, moschata (please check if this is correct)

Line 421                C. maxima, X C. moschata (please check if this is correct)

Line 475               Change Dnase to DNase

Line 480                “RNA CGMMV fragment” (need some correction)

Reviewer 2 Report

This manuscript introduces the multiplex PCR and Nanopore sequencing to detect the Cucumber green mottle mosaic virus, a vital virus under quarantine control in Australia. With a few exact figures, the article is easy to follow. However, many grammar mistakes and typos directly affect the manuscript's readability. Moreover, the significance of this work was not addressed in the current version. Below, please find a few suggestions.

Major:

1, In the abstract (Line 18), the authors proposed the challenges in virus detection, such as low virus copy number and low RNA quality. However, in the following part, these challenges were not addressed. Instead, the authors only claimed that "the method is easy to implement and makes HTS accessible to more labs." The authors should specify how the tiled amplicon method could be used to solve these challenges. Additionally, the abstract should be concise. A conclusion is missing after comparing the Cq values from different methods (Line 25 to 30).

2, Authors should clarify why CGMMV isolates VPRI43306 and NSW3-35 were chosen in this study and explain how the 13 primers were designed in the first section of the results.

3, The Discussion is too long and seems unorganized. I strongly suggest the authors add subtitles to mark different sections of the Discussion.

Minor:

1, please format the Ct and Cq though they stand for the same thing. Cq is not an abbreviation for the "cycle threshold."

2, Please pay attention to the grammar mistakes and typos in the manuscript. I only give a few examples. Line 308, "Minion" should be "MinION." Line 80, "More cost-effective." An objective is missing here. "Compared to …, Metagenomic HTS has been more…". Line 249, "that" should be "than"

3, Please format the reference list before formal submission.

Round 2

Reviewer 1 Report

Line 316: change “at least as sensitive than RT-qPCR” to “at least as sensitive as RT-qPCR”

Lines 334-336: Please make necessary correction in these lines to convey a clear message.

“CGMMV detection by RT-qPCR and TWG-Seq was not taken to below detectable levels with both tests able to detect one naturally infected seed in 1000 and nearly complete coding regions of CGMMV assembled by TWG-Seq.”

Line 392: Change 

“The use of controls such no-template” to “The use of controls such as no-template”

Author Response

Line 316: change “at least as sensitive than RT-qPCR” to “at least as sensitive as RT-qPCR”

Correction made 

Lines 334-336: Please make necessary correction in these lines to convey a clear message.

“CGMMV detection by RT-qPCR and TWG-Seq was not taken to below detectable levels with both tests able to detect one naturally infected seed in 1000 and nearly complete coding regions of CGMMV assembled by TWG-Seq.”

Section rewritten to improve clarity.

“Both RT-qPCR and TWG-Seq were able to detect CGMMV in a one naturally infected seed in 1000 seed dilution and a nearly complete coding region assembled by TWG-Seq. The detection limits of both methods were not reached, and future work could investigate the possibility of detecting one infected CGMMV seed in several 1000 seed and even if only partial genome information is acquired, TWG-Seq would provide greater confidence in detection compared to RT-qPCR. “

Line 392: Change 

“The use of controls such no-template” to “The use of controls such as no-template”

Correction made

Reviewer 2 Report

My concerns were addressed in the revised manuscript, except for the reference list. Please note that "formatted using Zotero MDPI citation style" cannot guarantee that the reference shares the same format. The authors should modify them one by one. For instance, some Journal names are in their full name style. While some are in abbreviations.

Author Response

My concerns were addressed in the revised manuscript, except for the reference list. Please note that "formatted using Zotero MDPI citation style" cannot guarantee that the reference shares the same format. The authors should modify them one by one. For instance, some Journal names are in their full name style. While some are in abbreviations.

References have been updated so that journal names are abbreviated.